# Climate Change Impacts on the Potential Distribution Pattern of *Osphya* (Coleoptera: Melandryidae), an Old but Small Beetle Group Distributed in the Northern Hemisphere

**DOI:** 10.3390/insects14050476

**Published:** 2023-05-18

**Authors:** Tong Liu, Haoyu Liu, Yongjie Wang, Yuxia Yang

**Affiliations:** 1The Key Laboratory of Zoological Systematics and Application, School of Life Science, Institute of Life Science and Green Development, Hebei University, Baoding 071002, China; liutong@stumail.hbu.edu.cn; 2Institute of Zoology, Guangdong Academy of Sciences, Guangzhou 510075, China; wangyjosmy@gmail.com

**Keywords:** *Osphya*, MaxEnt, geographical distribution pattern, global suitable areas, environmental factors

## Abstract

**Simple Summary:**

The spatial patterns of species are highly relevant to the contemporary environment and its topographical factors. It has been predicted that climate warming has the greatest impact on habitat selection and the expansion and contraction of geographic ranges for insects, as their physiological functions are strongly influenced by external environmental conditions. However, few studies focus on how the distribution pattern and range of the insects has or will change in response to long-term climate change. Under the background of climate warming, it is urgent for us to understand how the species distribution pattern is expected to change, in order to take some protection actions at the moment, particularly for those rare insect taxa. An old and Northern-Hemisphere-distributed beetle group *Osphya* is considered as an ideal candidate to conduct this aspect. In this study, we analyzed the distribution pattern and predicted the suitable habitats of *Osphya* under different climate scenarios (LGM, MID, Current, 2050s and 2070s) using ArcGIS techniques and MaxEnt modelling. The obtained results made us better understand how climate change affects the distribution pattern of the long-lived insects distributed in the Northern Hemisphere and provide guidance of the exploration and protection of the relict insects there such as *Osphya*.

**Abstract:**

Exploring the development of species distribution patterns under climate change is the basis of biogeography and macroecology. However, under the background of global climate change, few studies focus on how the distribution pattern and the range of insects have or will change in response to long-term climate change. An old but small, Northern-Hemisphere-distributed beetle group *Osphya* is an ideal subject to conduct the study in this aspect. Here, based on a comprehensive geographic dataset, we analyzed the global distribution pattern of *Osphya* using ArcGIS techniques, which declared a discontinuous and uneven distribution pattern across the USA, Europe, and Asia. Furthermore, we predicted the suitable habitats of *Osphya* under different climate scenarios via the MaxEnt model. The results showed that the high suitability areas were always concentrated in the European Mediterranean and the western coast of USA, while a low suitability exhibited in Asia. Moreover, by integrating the analyses of biogeography and habitat suitability, we inferred that the *Osphya* species conservatively prefer a warm, stable, and rainy climate, and they tend to expand towards higher latitude in response to the climate warming from the past to future. These results are helpful in exploring the species diversity and protection of *Osphya*.

## 1. Introduction

The exploration of the spatial distribution of species is considered as the basis and important part in biogeography and macroecology [1]. A lot of work on the spatial patterns of global species has been carried out over the past decades [2,3,4,5], which suggested that the spatial patterns of species are highly relevant to the contemporary environment and topographical factors such as geological history, climate stability, habitat heterogeneity, and climate change [6,7,8,9,10,11,12,13]. Among these factors, climate change plays a more important role not only in dominating the species spatial distribution patterns but also in altering species’ adaptive behaviors and survival conditions [14,15]. These effects are apparent in invertebrates, especially for insect groups, as their physiological functions are strongly influenced by external environmental conditions (e.g., temperature, humidity, and pressure) [16,17]. It has been supposed that climate warming is predicted to have the greatest impact on life-cycles, phenological patterns, habitat selection, and the expansion and contraction of geographic ranges for insects [18]. Meanwhile, it has been noted by the Intergovernmental Panel on Climate Change [19], with the increase of 1.5 °C (2 °C) in temperature, 6% (18%) of insects could lose half of their climatically suitable habitats. Under the background of climate warming, it is urgent for us to understand how the species distribution pattern will change as a response to climate change in order to take some protection actions at the moment, particularly for those rare insect taxa.

The genus *Osphya* Illiger, 1807, with low species diversity (a total of 28 species) [20] belonging to the subfamily Osphyinae of the family Melandryridae (Coleoptera: Tenebrionoidea), is restricted to Northern Hemisphere and is widely distributed in USA, Europe, and Asia. A previous study has dated its temporal origin as late as the mid-Cretaceous Era [21], which indicated that it had experienced a long evolutionary history. In addition, although the *Osphya* species is distributed globally, its occurrence is relatively high in the USA and Europe, while it is rare in Asia, however, where many areas are regarded as biodiversity hotspots in the world, such as in Southwestern China [14]. In the geological evolution of the Earth, the global climate has undergone long-term dramatic changes, which inevitably influence species distribution ranges and patterns [22]. Confronting the climate changes in geological history, how have the distribution pattern of the long-lived insect groups changed? There are very few studies focused on this aspect. Therefore, from a macroecological perspective, *Osphya* is considered to be an ideal subject to investigate the influences of long-term climate change on insect distribution patterns, owing to its long evolutionary history and a characteristic distribution scale.

Furthermore, the ongoing climate warming will considerably change local climate conditions, thereby altering the geographical distribution patterns of species, as organisms require favorable climate areas to maximize survival [23,24]. As an old and originally widely distributed group, *Osphya* has suffered drastic climate change over its long evolutionary history [8,21]; thus, it is speculated to have reduced in its group size and distribution range. This will be aggravated by the accelerated biodiversity crisis and biological extinction in the Anthropocene Epoch [25]. Given the narrow geographic distribution of each *Osphya* species, especially those from Asia, it seems plausible that habitat loss is causing many species to vanish without ever having been collected. However, there are still knowledge gaps in the geographical distribution and environmental adaptation of species during climate change. Thus, it is necessary for us to explore the impact of climate change both on species distribution patterns and habitat suitability via species distribution models (SDMs) to make some protection efforts to prevent ongoing loss or the extinction of the relict groups such as *Osphya*.

SDMs are statistically based models, which have been widely used to study autecology and the potential distribution of species under climate change [26,27,28]. SDMs can simulate the potential distribution of species across an area of interest by associating the observed distribution records (also known as presence records) of species with environmental factors [29,30,31,32], Recently, great advances have been made by means of SDMs in managing biological invasions [28,33,34,35], formulating biological conservation plans [14,36,37], and building global species distribution maps [38,39]. Several types of SDMs can be used for predicting the potential distribution of species, such as maximum entropy (MaxEnt) [40,41], Random Forest [42], boosted regress trees [43], generalized additive models [44], and so on. In the present study, the MaxEnt algorithm, which is a density estimation and species distribution prediction model based on the maximum entropy theory [45,46], will be introduced to construct the potential distribution patterns of *Osphya*. MaxEnt has been applied in realistic habitat simulations, the screening of eco-environment factors, and the quantitative descriptions of environmental factors [47]. Additionally, it has been successfully used to predict the potential distributions and environmental suitabilities of different groups, such as mammals [37], birds [48], amphibians [49], and insects [34,50,51,52]. Perhaps more importantly, we are going to apply MaxEnt for *Osphya* here because of its higher prediction accuracy and greater ability to analyze species with limited occurrence records and small sample sizes compared to other niche models [31,41,53,54,55,56,57].

In the present study, taking *Osphya* as the subject, we are going to analyze the characteristics of its distribution patterns, predict its potential distributions, and evaluate the important environmental factors affecting its distribution using ArcGIS and SDM techniques. Based on the obtained results, we aim to understand how the long-term climate change from the past to future affects the distribution patterns and ranges of the old and large-scale distributed insect group, from which we can receive some valuable information in better exploring the species’ diversity, making protection efforts to relict insects such as *Osphya*.

## 2. Materials and Methods

### 2.1. Distribution Data

Occurrence records of *Osphya* species worldwide were obtained from the Global Biodiversity Information Facility (GBIF, https://www.gbif.org/ (accessed on 15 December 2022)), California databases (SBMNH, http://www.sbcollections.org/ (accessed on 20 December 2022)), relevant literature records [58,59,60,61,62,63,64,65,66,67,68,69,70,71,72,73,74,75], and our field survey [20]. In total, 920 distribution records were collected, and they represent the current distribution of *Osphya*. Records that were incorrect or lacked geographical coordinates were proofread and supplemented by using Google Earth. We prepared and processed the distribution map of *Osphya* using ArcGIS 10.6 (ESRI Inc., California, LA, USA) and Adobe Photoshop 2020. In addition, we applied the “spatially rarefy occurrence data tools” in SDMToolbox with the resolution of 10 km to reduce the overfitting or incorrect predictions resulted from the spatial clustering of species records [76]. Finally, 298 valid occurrence records (Appendix A) were retained to construct the MaxEnt model.

### 2.2. Environmental Variables

In the present study, 19 bioclimatic factors and 1 topographic factor (Appendix A) were selected as initial modelling variables from the Coupled Model Intercomparison Project, Phase 5 (CMIP5) in WorldClim databases v1.4 (http://www.worldclim.org (accessed on 6 December 2022)), with a resolution of 2.5 min (~4.5 km). Bioclimatic factors in the current period (1960~1990), Last Glacial Maximum (LGM, approximately 22,000 years ago), and Mid-Holocene (MID, approximately 6000 years ago) were determined to predict the current and past potential distribution of the *Osphya* species. In addition, we also selected two types of Representative Concentration Pathways (RCP4.5 and RCP8.5) in the period of 2050s and 2070s as the proxies of future climate warming to predict the suitable area changes for *Osphya*. To reduce the impact of multicollinearity on the prediction process and improve the accuracy of model, we chose the variables whose contribution rate were higher than zero in the initial modelling for Pearson correlation analysis and variance inflation factor (VIF) by using R v4.0.3 (Appendix A) and removed the variables with high coefficients (|r| > 0.8) and VIF > 10 to raise the predictability of the model [77,78]. Consequently, 9 climate variables with no contribution rates or importance to the model’s prediction were excluded, and 11 variables (Figure 1) were retained to predict the final potential distribution of *Osphya*, including annual mean temperature (bio_1), mean diurnal range (bio_2), isothermality (bio_3), temperature seasonality (bio_4), mean temperature of wettest quarter (bio_8), mean temperature of driest quarter (bio_9), annual precipitation (bio_12), precipitation seasonality (bio_15), precipitation of driest quarter (bio_17), precipitation of coldest quarter (bio_19), and the elevation (ALT). In addition, the importance of each environmental variable was further determined based on the “Jackknife test” in MaxEnt v3.4.4 [41,79].

### 2.3. MaxEnt Modelling and Validation

MaxEnt version 3.4.4 (http://biodiversityinformatics.amnh.org/open_source/maxent/ (accessed on 20 May 2022)) [41,79] was applied to predict the potential distribution areas of *Osphya* based on 298 valid occurrence records and 11 predictor variables. All models set 75% of the distribution data as a train set and the remaining 25% as a test set, with 10 replicates cross-validated as a replicated run type, and the iterations of the modelling were set from 500 to 5000 to allow model to have adequate time for convergence [80]. Other parameters were kept as the default. Additionally, it was required that the species should be restricted to a similar ecological niche in the prediction by MaxEnt [81]. Thus, the species with more than five distribution records were predicted alone, while others were together. The seven final SDMs’ suitability logistic outputs were then stacked to produce the potential distribution of *Osphya*.

The performance of the model was measured by the receiver operating characteristic curve (ROC), and the precision was further calculated using the area under the ROC curve (AUC), an index to evaluate the accuracy of simulation [34,35,48,82]. The theoretical value range of the AUC was from 0.5 to 1.0, and AUC values were closer to 1.0 for a higher accuracy of the model [33]. Although AUC has been widely used to evaluate the prediction accuracy, it remains controversial for some associated disadvantages [83,84]. To improve model fitting, we additionally selected the true skill statistic (TSS) as the associate measured indictors to evaluate the performance of SDMs [85]. The TSS value ranged from −1 to +1, and values between 0 and −1 indicated performance no better than random [85].

### 2.4. Dynamic Change in Suitable Areas

To explore the changes in the potential distribution areas of *Osphya* under different climate scenarios, we first defined suitable/unsuitable areas based on “the average training presence threshold” obtained by MaxEnt models. Then, we used the “Distribution Changes Between Binary SDMs” function in SDMtoolbox v2.4 to simulate the changes in the suitable area of *Osphya* under different climate scenarios [86] and, finally, obtained two types of outputs. One is a table containing statistical information relevant to the species’ ranges changes. The other was a spatial map showing the gains or losses of suitable areas for candidates in different climate scenarios, which were defined as four types as “range contraction”, “range expansion”, “no change”, and “no occupancy”, respectively.

## 3. Results

### 3.1. Geographical Distribution Pattern of Osphya

The *Osphya* species are distributed within the geographical range 9.99~17.60° N, 121.65~127.78° E of the Northern Hemisphere, including the USA, the Mediterranean region of Europe, and West, South, and East Asia (Figure 2).

The highest species richness of *Osphya* is found in the Mediterranean region of Europe (including the westernmost Asia), with nine species distributed there, accounting for 32% of the total number. The members of *Osphya* in USA are also abundant, with six species separated by the Rocky Mountains in three isolated regions: the western coast, the south end, and the eastern region. The remaining species (a total of 13 species) are scattered in South and East Asia, and each species is narrowly ranged and restricted to a single locality.

### 3.2. Potential Distribution of Osphya in Current Period

The AUC and TSS values obtained by the MaxEnt models for *Osphya* under the current climate were 0.967 and 0.899, respectively (Appendix A), indicating the model’s high performance and discrimination power for the current potential distribution of *Osphya*. The potential distribution range (Figure 3) was almost congruent with the current distribution pattern (Figure 2), except for the predicted areas in the Southern Hemisphere. The potentially suitable area of *Osphya* covered 1049.21 × 10^4^ km^2^ of the world’s scale (Appendix A), and those in higher suitability (high and medium) were mainly centralized in the areas around the Mediterranean Sea of Europe and the coastal areas of the USA (southwest coasts), covering a total area of 33.65 × 10^4^ km^2^. In Asia, only a few small areas were predicted in general suitability, such as northeast Afghanistan and southwest India. Although a relatively large area of southern China was predicted within the potential range; it was mostly of low suitability (Figure 3).

### 3.3. Potential Distribution of Osphya under Past and Future Climate Scenarios

The MaxEnt models showed high accuracy for *Osphya* under the past and future climates, with AUC and TSS scores greater than 0.95 and 0.85, respectively (Appendix A), which indicated the high discrimination of the past and future potential distributions for *Osphya*. The potential distribution patterns of *Osphya* during LGM (Figure 4A) and MID (Figure 4B) were similar to that of the current period (Figure 3), which generally exhibited an expanding trend towards a higher latitude from the past to the present but never beyond the mid-latitude of the Northern Hemisphere. The global suitable area of *Osphya* under LGM and MID periods covered 645.98 × 10^4^ km^2^ and 1015.18 × 10^4^ km^2^, respectively (Appendix A). The total potential distribution range of LGM was smallest (Figure 4A), which shrunk evidently in the Mediterranean area and the USA, which was completely absent in Japan. Similar conditions also occurred in China, where the suitable habitats were much smaller than those of the current (Figure 4A). However, the high-suitability area was largest (4.56 × 10^4^ km^2^) under the LGM scenario and concentrated around the Mediterranean Sea (Appendix A and Figure 4A). In comparison, although the total suitable area of *Osphya* increased during the MID period, its highly suitable area decreased significantly, covering only 0.42 × 10^4^ km^2^ (Appendix A and Figure 4B).

Compared with the current period, the suitable area of *Osphya* would significantly increase under the future climate scenarios (2050s RCP4.5: 1051.73 × 10^4^ km^2^; 2050s RCP8.5: 1257.49 × 10^4^ km^2^; 2070s RCP4.5: 1163.56 × 10^4^ km^2^; 2070s RCP8.5: 1196.31 × 10^4^ km^2^), whereas the high-suitability area would constantly decrease (corresponding to 1.22 × 10^4^ km^2^; 2.18 × 10^4^ km^2^; 1.48 × 10^4^ km^2^; 2.00 × 10^4^ km^2^; respectively) (Appendix A and Figure 5). Similar to the changes from the past to current periods, the overall range of suitable areas for *Osphya* would continue move northwards in the future projections (Figure 4C–F).

### 3.4. Changes in Suitable Areas of Osphya from Past to Future

The relative changes in *Osphya* were obtained by comparing the potential distribution areas among the current and past/future climate scenarios. The results showed that the expansion area of *Osphya* was always larger than its contraction area under different climate scenarios (Appendix A and Figure 6), which indicated that the suitable area of *Osphya* would constantly increase from past to future. During the period from the LGM to the present, significant range expansions of the suitable areas for *Osphya* mainly occurred in the parts of the Central and North West USA, France, Britain, southern and northwestern Iran, northern Afghanistan, South China, and south Japan. In comparison, during the period from the present to the future, expansion would occur in the areas of the Central and North West USA, southeast Greenland, Svalbard Islands, Central and Northern Norway, and sporadic areas in Central and East Asia (Figure 6).

### 3.5. Determinants Affecting Geographical Distribution of Osphya

In this study, eleven environmental variables used for the predictive models were evaluated by the “Jackknife test” procedure using MaxEnt, and five environmental factors were finally determined as the most important factors affecting the potential distribution of *Osphya* (Appendix A), including the annual mean temperature (BIO1; 19.3% of contribution), isothermality (BIO3; 16.3% of contribution), mean temperature of the driest quarter (BIO9; 1.9% of contribution), temperature seasonality (BIO4; 14% of contribution), and the precipitation of the coldest quarter (BIO19; 32.6% of contribution) (Appendix A).

The response curves from the MaxEnt output depicted the variations in the logistic values imparted by the changes in each predictor when all other variables remained at their average values [48]. Based on the specific existence thresholds (>0.2), we found that *Osphya* preferred habitats with an annual mean temperature (bio_1, Appendix A) of more than 23.1 °C; an isothermality (bio_3, Appendix A) ranging from 0.20 to 0.55, of which 0.33 was the best; a temperature seasonality (bio_4, Appendix A) of less than 90.7, of which 62.9 was the best; a mean temperature of the driest quarter (bio_9, Appendix A) ranging from 4.8 °C to 23.4 °C, of which 16.0 °C was the best; and the precipitation of the coldest quarter (bio_19, Appendix A) ranging from 31.4 mm to 563.8 mm, of which 117.4 mm was the best.

## 4. Discussion

### 4.1. Characteristics and Formation of Distribution Pattern of Osphya

In general, all the *Osphya* species occurred in the Northern Hemisphere, within the latitude zone ranging from 20° N to 35° N, corresponding to the temperate and subtropical regions. The overall distribution pattern of *Osphya* was discontinuous and uneven (Figure 2), which may have been caused by the joint effects of the geological history and environmental adaptions [87].

Clearly, no species of *Osphya* have been found in the Southern Hemisphere, which may have been closely related to their spatial origins and evolutionary histories. According to the present distribution patterns, *Osphya* is supposed to be of a Laurasian origin after the permanent separation of Laurasia and Gondwana (approximately in 174 Mya) [88], which occurred in the Mesozoic Era. This was consistent with the estimated origin time of *Osphya* (ca. 105.7 Mya, mid-Cretaceous period) [21]. At that time, Laurasia was an integrated land without the huge straits impeding species dispersal [89], which made the possibility for the *Osphya* species to widely spread in the USA and Eurasia. Subsequently, with the opening of the Atlantic Ocean during the Cretaceous period, the lands of the East and West Atlantic were separated, which led to the formation of the geographical patterns of a transoceanic distribution [90], as was the case for *Osphya*. However, frequent movements of continental plates and islands restricted the further dispersal of organisms into the interior of the continent during the Cretaceous and Cenozoic Eras [90], given the fact that some biotas exhibited shifts in biodiversity from the western Tethyan region (Mediterranean Sea) to the Indo-West Pacific during the past 50 million years [91,92]. Thus, within Eurasia, the *Osphya* species was presumed to have spread from west to east along the north coast of the Tethys Sea. Later, the collision between the Indian and the Eurasian plates occurred in the Cenozoic Era, which led to the convergence, contraction, and complete closure of the Tethys Sea [93,94,95] and the formation of the Himalayas at approximately 50 Mya during the Early Eocene period [96,97]. This was congruent with the distribution pattern of *Osphya* along the Himalayas and South East Asia. From then on, the global distribution patterns of the *Osphya* species were preliminarily formed.

In addition, the species richness of *Osphya* was much higher in Western Europe (Mediterranean region) and the USA (western and central regions), while the species were sparsely scattered in West, South, and East Asia, showing a relatively uneven distribution pattern. Similar distribution patterns have also been revealed in other organisms, such as plants [98,99], millipedes [100,101], spiders [102], and insects [103,104]. As is well known, the Mediterranean region is an area with a high species diversity [105] and has been recognized as one of the first global biodiversity hotspots [2] due to its unique conjunction of geography, history, and climate conditions [106], which provides suitable habitat conditions for the speciation and diversification of *Osphya* species in the region. Meanwhile, Western and Central USA (e.g., California, Arizona, and Texas) are the important biodiversity centers possessing thousands of plants and animals in the world [107]. The complex topographies (e.g., high elevation mountain ranges and the Channel Islands), various landscapes (e.g., lush coastal coniferous forests and alpine tundra), and suitable oceanic climates in these regions drive the formation of diverse habitats, which provides more opportunities for the speciation, evolution, and coexistence of a variety of unique species [108], and *Osphya* is one of the beneficiaries.

By contrast, although nearly half the species of *Osphya* exist in Asia, the species richness is quite low in this region. In other words, they are rarely found in forested habitats and remain quite mysterious due to their small populations. Furthermore, China covers a vast geographical area with diverse landscapes, and its southwestern area is recognized as one of the most globally biodiverse hotspots [109], but only one *Osphya* species was recorded from Mainland China, except from two species from the islands of Taiwan. Perhaps fewer distribution records of *Osphya* in these regions indicate a relative lack of collection efforts, but it is more likely that this reflects a reasonable distribution pattern when considering the biogeographical history (discussed as above) and environmental factors. Similar to those of Europe and the USA, the Asian *Osphya* species are mainly concentrated on coastal regions (southwest coast of India and Myanmar) and islands (e.g., islands of Taiwan, Honshu Island) that maintain an oceanic climate. Previous studies have proved that islands play an important role in long-term speciation and maintenance due to their relatively slow biotic turnovers and strong thermal stabilities [3,5]. The Taiwan and Japan areas are mountainous islands with mountain ranges running north–south throughout the island, which result in high heterogeneities of their ecological environments, thereby providing diverse habitats for the survival of endemic biota [110,111]. Additionally, the two islands are located at the easternmost Asian mainland, and they are continental islands. Thus, the formation of land bridges between them and the continent [3,111,112,113] guaranteed the communication of the species despite the paleoclimate changes [114,115]. However, the two island habitats have low suitability, probably due to the influence of the hot or unstable monsoon climate [116]. In addition, a few species of *Osphya* also occur in the montane areas of the inland regions of Asian countries (e.g., Himalayas, Shennongjia, Wuyi Mountains), probably owing to the complex terrains and high habitat heterogeneities in these regions [3,110,111]. The complex topographies and diverse environments in these areas result in different altitude gradient climates and habitats [3,117], which have accelerated niche differentiations and provided diverse habitats for the survival of species in these regions [14]. Particularly, the stable climates in these areas during the glacial periods might have been the refugia for the organisms in favor of maintaining relict species [118], such as *Osphya*, which has a long evolutionary history and a once-wide distribution across the Laurasian Land.

### 4.2. Dynamic Changes in Potential Distribution of Osphya

In terms of our findings, based on the MaxEnt model, we investigated the potential distribution areas under different climate scenarios. The results showed that the suitable areas of *Osphya* under the current climate (Figure 3) were widely distributed in the mid-latitude of the Northern Hemisphere, including Western and Central USA and parts of Europe and Asia, which were consistent with its geographical distribution patterns (Figure 2). Compared with that of the current situation, the potential distribution range greatly shrank in the LGM period, especially in the Central USA and Mediterranean regions (Figure 4A), which may have been closely associated with the geological events at that time. There used to be huge ice sheets covering large parts of the North America, Europe, and North Asia during the LGM period (e.g., the Laurentide and Innuitian ice sheet) [119], which led to the relatively cold and dry habitats, restricting the survival of biotas in these areas [120,121]. Thus, organisms inhabiting these regions were forced to search for glacial refugia, such as montane valleys, mountainous islands, or terrestrial wet-spots [3,5,14,122]. Glacial refugia, undeniably, were critical for the long-term survival and dynamics of temperate biodiversity during the LGM period [123]. This was particularly reflected in the Mediterranean Sea region, where the habitat area with high-suitability was 4.56 × 10^4^ km^2^ for the *Osphya* species under the LGM scenario. During glaciations, islands in this area served as refuges [124,125], and, subsequently, during the deglaciation and warming, they became the centers of species prosperities [120]. Therefore, the Mediterranean Sea region of Europe could provide highly suitable habitats for the *Osphya* species, no matter how harsh or variable the climate was during the LGM period.

With the temperature increasing from the past (LGM and MID) to future (2050s RCP4.5/8.5 and 2070s RCP4.5/8.5), the total area of the suitable habitat of *Osphya* has constantly extended, especially more clearly shown in Europe and Greenland, which may have resulted from the effects of global warming. Under the background of climate warming, the restrictions caused by low temperatures on the survival and development for insects distributed in high latitude regions have been alleviated, thereby leading to the continuous expansion of their suitable areas [126]. However, the medium-high suitability area for *Osphya* has significantly decreased, especially in the subtropical area near to the Equator (Figure 4), where the temperature is probably too hot for the *Osphya* species inhabiting there, given that they may prefer relatively warm and humid climatic conditions (see discussion in Section 4.3). In addition, climate warming increases the chance for insects restricted by lower temperatures to spread towards higher latitudes; thus, the suitable areas of the *Osphya* species have or will continue to spread northwards worldwide, and the lower latitude areas will become unsuitable for their occurrence, such as the South China and the Indo-Chinese regions.

Above all, the potential distribution patterns of *Osphya* in different periods are almost consistent, which suggests that the *Osphya* species probably inhabit similar habitats in their evolutionary histories. Although some regions of the Southern Hemisphere (such as South Africa and Australia) were predicted as potentially suitable areas for *Osphya* under different climate scenarios, no *Osphya* species have been found there, which may be theoretically relevant to their spatial origins in the Northern Hemisphere. In addition, with the continuous warming of the climate from the past to future, the suitable area of *Osphya* constantly increases worldwide, which indicates that the current distribution is far from reaching saturation, and the species still have a great probability of being discovered in the future. Thus, it is necessary to conduct a large-scale survey in the areas with potential distribution probabilities to investigate the diversities and quantities of the species in each area, which will provide a scientific basis for understanding the species diversities of *Osphya*. Particularly, as a relict group of beetles, *Osphya* species also play an important role in understanding the evolution of Melandryidae or even the higher level of Coleoptera. Thus, it is urgent and necessary for its conservation. In terms of the obtained results in the present study, we should pay much attention to the potentially suitable areas for *Osphya*. On the one hand, we could receive some guidance in discovering the hitherto unknown species. On the other hand, we call for protection efforts in these areas, which are probably critical in the formation and maintenance of biodiversity.

### 4.3. Environmental Variables Affecting Suitability of Osphya

The results obtained by the Jackknife test demonstrated that the *Osphya* species were more likely to live in warm, humid, and stable habitats. Furthermore, combined with the predicted suitable distribution range, we concluded that the *Osphya* species preferred inhabiting areas with oceanic climates, such as the Mediterranean area of Europe and the coastal areas of the USA, where the climates are stable, warm, and rainy [127,128]. This was consistent with aforementioned results of the Jackknife test.

All the important factors that affect the potential distribution and survival of *Osphya* are closely related to precipitation and temperature. It is suggested that water–energy factors dominate the formation of insect diversity, and inappropriate water–energy can exert a significant impact insect distribution, morphology, phenology, and even survival [109,129]. However, the bionomics of *Osphya* were actually very little known. Only Nikitsky (1992) stated that the larvae of *O. orientalis* [60,130] live in rotten dead wood, perhaps also in the soil. Precipitation will cause changes in soil moisture [131]. Thus, it is supposed that excessively dry or wet soil is detrimental to the development of *Osphya*. Except for precipitation, the remaining important factors affecting the occurrence probabilities of *Osphya* are all correlated with the temperature, which is also considered one of the main factors limiting the distribution of insects on Earth [132]. The temperature change will not only affect the external habitats of organisms but will also pose an important challenge to the intrinsic mechanism of organisms, especially in insects [133,134], as their basic physiological functions are strongly influenced by external temperatures [16]. Previous studies have shown that the temperature rises caused by climate change would increase the risk of sudden and severe biodiversity losses, especially in the tropics [135,136,137]. Therefore, the suitable habitats of many insects will gradually migrate to higher latitude regions with global warming, such as Odonata [133] and butterflies [134], as is the case for *Osphya.*

### 4.4. Limitations

Any attempts based on the SDMs inevitably meet several limitations and biases, and our study was no exception. First of all, the evaluation metrics of the SDMs merely represented the quality of the simulation rather than the actual distribution of the candidates, which indicated that the results obtained via the SDMs were probably phenomenological and not mechanistic [82]. In addition, although we included many environmental variables when building the SDMs, they were still insufficient to determine the true distribution conditions of the species because many other important variables were missing from our models, such as biological interactions [138], topography constraints [5], human activities [139], and natural enemies [140]. All of these factors play a significant role in their distribution locations and ranges, and, even under suitable biological and abiotic conditions, species may still lose their suitable distribution areas for some time [141,142]. Moreover, the assumption that the current distribution of candidates was in equilibrium with the state under different climate scenarios was unreasonable [143] because the distribution of candidate populations was dynamic rather than static.

Another limitation stems from the data bias. Model building usually requires solely presence-only and pseudo-absence data, which merely reflects the observation density used to train model data rather than the causal mechanism of the actual distribution of the species [143]. Additionally, the distribution data of candidates used for modelling have different temporal scales, which indicate that the occurrence records used for modelling are “historical” rather than “real-time”. Therefore, we cannot determine whether results obtained by SDMs using bias data are reliable or not. In addition, a taxonomists’ specialization of taxon studies may be another potentially important factor affecting the data integrity, especially for those rare or endemic groups, which will inevitably enlarge the data bias and impede the building of SDMs [109].

Although there are several defects in technology and species data, we consider that the obtained results in this study are helpful for us in the discovery and protection of *Osphya*. With the increase in various data, including species distribution records, biological requirements, and ecological requirements, etc., together with the rapid developments of SDMs, we believe that a better prediction will be worked out to verify the present results.

## 5. Conclusions

A comprehensive geographic distribution dataset including all hitherto known *Osphya* species was compiled and analyzed by ArcGIS and MaxEnt techniques, and the spatial distribution patterns and potentially suitable areas were obtained. As a result, a discontinuous and unbalanced distribution pattern of *Osphya* was presented, with a transoceanic distribution in the USA and Eurasia, where the species richness was much higher in the USA and Western Europe than in Asia. The formation of this distribution pattern was discussed and presumed to be relevant to its geological history and environmental adaptations. Furthermore, the environmental factors affecting the suitability of *Osphya* were evaluated, and the results demonstrated that the *Osphya* species conservatively preferred a warm, stable, and rainy climate, and their highly suitable areas were mainly located in parts of the Mediterranean area of Europe and the coastal areas of USA. Moreover, the analyses of dynamic changes in the potential distribution showed that the total area of the suitable habitat of *Osphya* has or would constantly extend from the past to future, and the distribution range would spread northwards worldwide in response to global warming. These results made us understand how long-term climate change from the past to future affect the distribution patterns and ranges of *Osphya*, which give us some guidance in exploring the species diversities of *Osphya* in the field trip, meanwhile providing some theoretical basis for our protection efforts exerted to this relict beetle group in the future. Nevertheless, a better prediction produced with more basic analysis data and the updating of SDMs are required in future to verify the obtained results in the present study.

## Figures and Tables

**Figure 1 insects-14-00476-f001:**
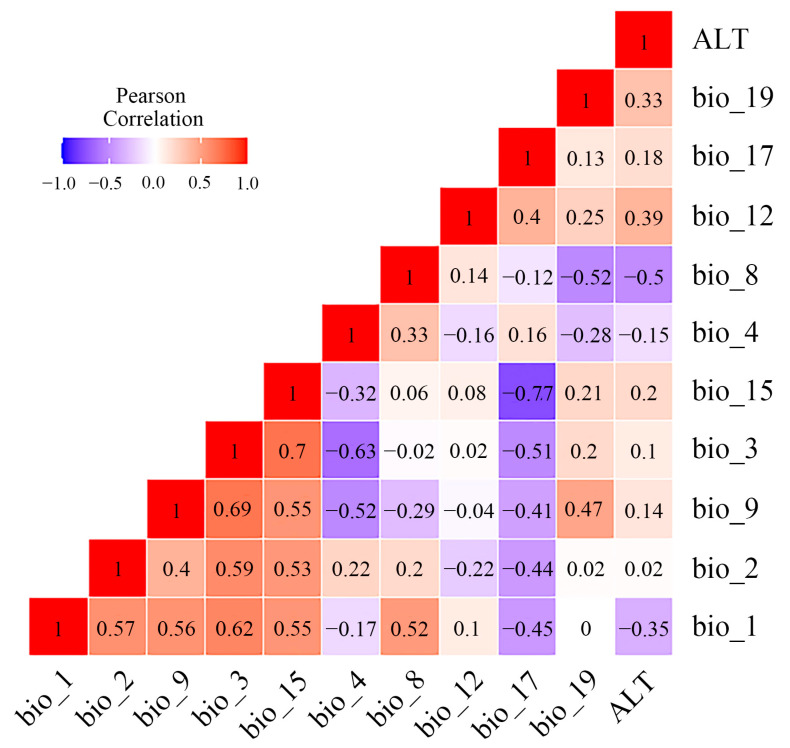
Multicollinearity test of Pearson correlation analysis between environmental variables retained for MaxEnt modeling.

**Figure 2 insects-14-00476-f002:**
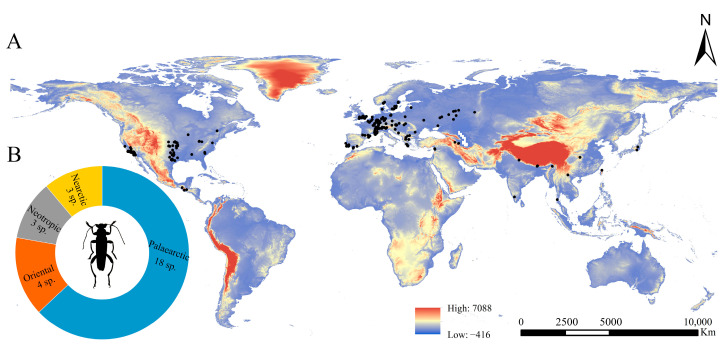
(**A**) Distribution map of *Osphya* species in the world; (**B**) numbers of *Osphya* species located in each zoogeographical region.

**Figure 3 insects-14-00476-f003:**
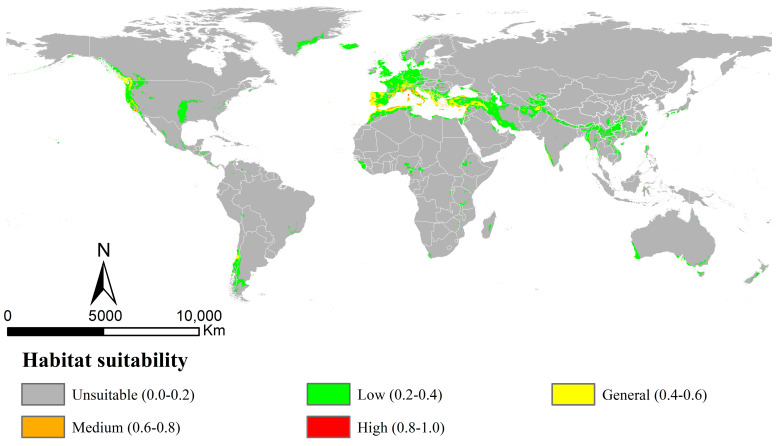
Potential distribution of *Osphya* in current period (1960~1990) using MaxEnt with bioclimatic factors.

**Figure 4 insects-14-00476-f004:**
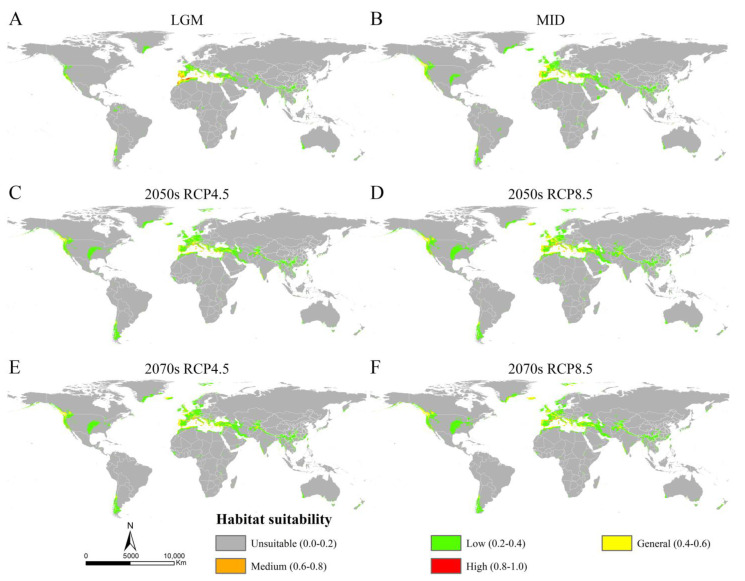
The potential distribution areas of *Osphya* under different climate scenarios. (**A**) Last Glacial Maximum (LGM, ca. 22 ka); (**B**) Mid-Holocene (ca. 6 ka); (**C**) 2050s RCP4.5; (**D**) 2050s RCP8.5; (**E**) 2070s RCP4.5; (**F**) 2070s RCP8.5.

**Figure 5 insects-14-00476-f005:**
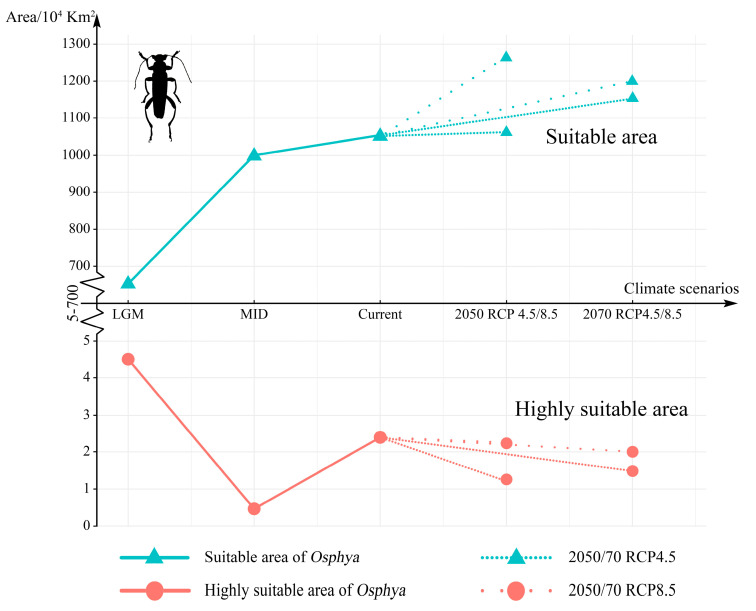
Changes in the suitable and highly suitable area of *Osphya* under different climate scenarios. The indigo-blue triangle line in the upper represents the change in suitable area, while the pink-circle line in the lower is the change in highly suitable area. The high-density line segments represent 2050/70 RCP 4.5 climate scenarios, while the low-density line segments are 2050/70 RCP 8.5.

**Figure 6 insects-14-00476-f006:**
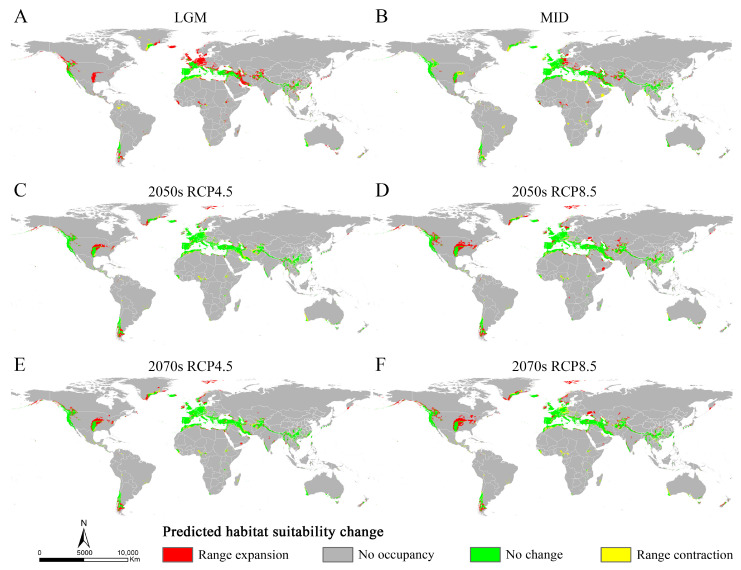
Changes in distribution areas of *Osphya* under different climate scenarios based on MaxEnt model. (**A**) Last Glacial Maximum (LGM, ca. 22 ka); (**B**) Mid-Holocene (ca. 6 ka); (**C**) 2050s RCP4.5; (**D**) 2050s RCP8.5; (**E**) 2070s RCP4.5; (**F**) 2070s RCP8.5.

## Data Availability

Data used in the analyses are available at Appendix A.

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
