# Peer review of "Climate Change Impacts on the Potential Distribution Pattern of Osphya (Coleoptera: Melandryidae), an Old but Small Beetle Group Distributed in the Northern Hemisphere"

_insects, 2023, doi:10.3390/insects14050476_

Round 1

Reviewer 1 Report

The MS is finely prepared. I recommend to accept it for publication. 

But the group studied is too few species and the paper may not provide strong support for the conclusion obtained. 

Author Response

RESPONSE TO REVIEW COMMENTS

Manuscript Title: Climate change impacts on the potential distribution pattern of Osphya (Coleoptera: Melandryidae), an old but small beetle group distributed in the North Hemisphere

REVIEWER #1:

Q: The MS is finely prepared. I recommend to accept it for publication.

Response: We thank the reviewer for reading our paper carefully and giving the above positive comments.

Q: But the group studied is too few species and the paper may not provide strong support for the conclusion obtained.

Response: Thank you for your valuable comments. It is true that Osphya is a small beetle group, but the simulation of SDMs requires the occurrence data (≥5 records) for each species rather than the number of species (Brown et al., 2014; Chen et al., 2017), and our study meets this criterion. Although there are some deficiencies noted in the discussion part, we believe that our findings and conclusions obtained in this study by using the distribution data of Osphya and SDMs are relatively objective. Thanks a lot.

Additional references:

Brown, J.L.; Cameron, A.; Yoder, A.D.; Vences, M. A necessarily complex model to explain the biogeography of the amphibians and reptiles of Madagascar. Nat. Commun. 2014, 5, 5046. doi: 10.1038/ncomms6046.

Chen, Y.H; Zhang, J.; Jiang, J.P.; Nielsen, S.E.; He, F.L. Assessing the effectiveness of China's protected areas to conserve current and future amphibian diversity. Divers. Distrib. 2017, 23, 146–157. doi: 10.1111/ddi.12508.

Reviewer 2 Report

This is an interesting study that models the current and potential changes in distribution of members in the genus Osphya with a warming climate. The study used ArcGIS and MaxEnt to model the change in the genus distribution by selecting the best climatic variables and analysing the model output under different climate scenarios. It concluded that this genus prefers a warm, humid, stable, and rainy climate and the most preferred habitat for this genus will become less suitable with climate warming.

The introduction is thorough in explaining the context for undertaking the modelling and leads to the approach taken for the study. Methods are given in detail and the rationale for the reasons why variables were chosen and or excluded along with steps to avoid bias and improve model fit are explained well.

Specific comments

Introduction

The importance of studying this genus under future conditions needs to be expanded in the introduction. Is its habitat under threat from global warming or is the genus becoming more impacted by human populations and industry? Why should we be concerned for its ecology? It might be a species poor genus though it is widely distributed and as such the importance of studying this genus should be outlined. Otherwise, one could choose any widely distributed genus in the Northern Hemisphere and undertake an SDM analysis.

 Results

Figure 2. Increase the font size of Figure 2B

Figure 4. Increase the font size of the legend.

Figure 5. This figure needs a lot of work as it is unclear which results are being presented.

Firstly, please insert the “Area/10^4 Km^2” next to the values of the axis and write change on the lower axis. Please write next to the legend icons what they mean and place the legend across the bottom of the figure. Finally change the intensity of the dashes for the low and high density to make each climate model very clear in the graph.

Line 265 Please insert and explain if not present the ‘Jackknife test’ in the relevant section of the method as I could not find this written in the methods section.

Discussion

Line 282 change “the Osphya species are all occurring” to “all the Osphya species occurred”

There are several spelling and grammatical errors throughout this manuscript. Please check the manuscript for similar errors as it makes it hard to undersrtnad certain sections of the mansuscript

Specific comments

Lines 25-28. Change to “Exploring the development of species distribution pattern under climate change forms the basis of and plays an important part in the mechanisms within biogeography and macroecology”

Lines 54-58. “has significantly potential” change to “is predicted to have the greatest impact” and “Intergovernmental on Climate Change” add the word “Panel” before “on”.

Line 62 “a species-poor beetle group” change to “with low species diversity”

Line 76 change “gain” to “maximise”

Line 98 change “better deal” to “greater ability to analyse species”

Line 111 change “world range” to “globally” or “worldwide”

Line 113 should this be “field” not “filed” survey. Please check if this type of error may be occurring throughout the manuscript.

Line 115 change “Records data if” to “Records that were”

Line 166 change “Given this situation” to “To improve model fitting”

Line 326 change “field trip” to “native habitat” or forested habitat”

Line 334 change “coast to “coastal”

Line 335 change “where is the oceanic climate” to “that maintain an oceanic climate”

Line 340 change “the biota survivals” to “the survival of endemic biota”

Line 346 change “inland” to something like “Asia” or “inland regions of Asian countries”

Line 486 add ‘the” before “future”

Author Response

RESPONSE TO REVIEW COMMENTS

Manuscript Title: Climate change impacts on the potential distribution pattern of Osphya (Coleoptera: Melandryidae), an old but small beetle group distributed in the North Hemisphere

REVIEWER #2:

Q: This is an interesting study that models the current and potential changes in distribution of members in the genus Osphya with a warming climate. The study used ArcGIS and MaxEnt to model the change in the genus distribution by selecting the best climatic variables and analyzing the model output under different climate scenarios. It concluded that this genus prefers a warm, humid, stable, and rainy climate and the most preferred habitat for this genus will become less suitable with climate warming.

Response: We are very grateful to the reviewers for their precious time in reviewing this manuscript and making positive comments.

Q: The introduction is thorough in explaining the context for undertaking the modelling and leads to the approach taken for the study. Methods are given in detail and the rationale for the reasons why variables were chosen and or excluded along with steps to avoid bias and improve model fit are explained well.

Response: Thanks for your positive comments.

Specific comments

Introduction

Q: The importance of studying this genus under future conditions needs to be expanded in the introduction. Is its habitat under threat from global warming or is the genus becoming more impacted by human populations and industry? Why should we be concerned for its ecology? It might be a species poor genus though it is widely distributed and as such the importance of studying this genus should be outlined. Otherwise, one could choose any widely distributed genus in the Northern Hemisphere and undertake an SDM analysis.

Response: Thanks for your valuable comments. Following your suggestion, we have made some supplementary explanations in the introduction part. Please see the details in the revision manuscript.

Results

Q: Figure 2. Increase the font size of Figure 2B.

Response: Thanks for your rigorous suggestions. We have revised it according to your suggestions.

Q: Figure 4. Increase the font size of the legend.

Response: Thanks very much for your comments, accordingly, we have revised it.

Q: Figure 5. This figure needs a lot of work as it is unclear which results are being presented. Firstly, please insert the “Area/10^4 Km^2” next to the values of the axis and write change on the lower axis. Please write next to the legend icons what they mean and place the legend across the bottom of the figure. Finally change the intensity of the dashes for the low and high density to make each climate model very clear in the graph.

Response: Thanks for your suggestions, and accordingly we made changes.  

Q: Line 265 Please insert and explain if not present the ‘Jackknife test’ in the relevant section of the method as I could not find this written in the methods section.

Response: Thanks for your valuable comments. We have added relevant content in the “method” section, please see the details in the revision manuscript.

Discussion:

Q: Line 282 change “the Osphya species are all occurring” to “all the Osphya species occurred”

Response: Thanks for your suggestions, and accordingly we made changes.

Comments on the Quality of English Language: There are several spelling and grammatical errors throughout this manuscript. Please check the manuscript for similar errors as it makes it hard to understand certain sections of the manuscript.

Response: We are very sorry to confuse you due to our inaccurate expression. We have revised the corresponding parts, please see the detail in the revision manuscript.

Specific comments

Q: Lines 25-28. Change to “Exploring the development of species distribution pattern under climate change forms the basis of and plays an important part in the mechanisms within biogeography and macroecology”

Response: Thanks for your valuable comments. We have revised it according to your suggestion.

Q: Lines 54-58. “has significantly potential” change to “is predicted to have the greatest impact” and “Intergovernmental on Climate Change” add the word “Panel” before “on”.

Response: Thanks for your suggestions, and accordingly we have revised it.

Q: Line 62 “a species-poor beetle group” change to “with low species diversity”

Response: Thanks for your suggestion. We have revised it in the revision manuscript.

Q: Line 76 change “gain” to “maximise”

Response: Thanks for your rigorous comment. We have revised it according to your suggestion.

Q: Line 98 change “better deal” to “greater ability to analyse species”

Response: Thank you for your comments. We have revised it according to your suggestion.

Q: Line 111 change “world range” to “globally” or “worldwide”

Response: Thanks for your comment. We have revised it according to your suggestion.

Q: Line 113 should this be “field” not “filed” survey. Please check if this type of error may be occurring throughout the manuscript.

Response: Thanks for your suggestions, and accordingly we have revised it.

Q: Line 115 change “Records data if” to “Records that were”

Response: Thanks for your suggestions, and accordingly we have revised it.

Q: Line 166 change “Given this situation” to “To improve model fitting”

Response: Thanks for your suggestion. We have revised it in the revision manuscript.

Q: Line 326 change “field trip” to “native habitat” or forested habitat”

Response: Thanks for your comment. We have revised it according to your suggestion.

Q: Line 334 change “coast to “coastal”

Response: Thanks for your comment, and accordingly we have revised it.

Q: Line 335 change “where is the oceanic climate” to “that maintain an oceanic climate”

Response: Thanks for your suggestion. We have revised it in the revision manuscript.

Q: Line 340 change “the biota survivals” to “the survival of endemic biota”

Response: Thanks for your comment, and accordingly we have revised it.

Q: Line 346 change “inland” to something like “Asia” or “inland regions of Asian countries”

Response: Thanks for your suggestion. We have revised it in the revision manuscript.

Q: Line 486 add ‘the” before “future”

Response: Thanks for your comment, and accordingly we have revised it.

Reviewer 3 Report

1. A simple summary and an abstract are not much different. It is necessary to improve the abstract.

2. Line 78-80. Unnecessary information. The authors at the end of the introduction define the purpose and objectives of the study.

3. In the results of the research, the authors did not take into account publications and data from Russia and Ukraine. In particular, Osphya bipunctata (Fabricius, 1775) lives in these countries and information about the area is available on the platform https://www.gbif.org/species/4453653. See publications (https://doi.org/10.3390/d14121128; https://doi.org/10.3390/d14100825). You need to add these publications to the references.

4. Some captions to the drawings have been moved to other pages, which makes it difficult to read the manuscript.

5. In the discussion, due to the lack of all publications for analysis, the authors are mistaken in their conclusions. This requires making changes to the text.

Author Response

RESPONSE TO REVIEW COMMENTS

Manuscript Title: Climate change impacts on the potential distribution pattern of Osphya (Coleoptera: Melandryidae), an old but small beetle group distributed in the North Hemisphere

REVIEWER #3:

Q: 1. A simple summary and an abstract are not much different. It is necessary to improve the abstract.

Response: Thanks for your valuable comment. We have revised it in the main text, please see the details in the revision manuscript.

Q: 2. Line 78-80. Unnecessary information. The authors at the end of the introduction define the purpose and objectives of the study.

Response: Thanks for your valuable comments. We have revised it in the main text according your suggestions.

Q: 3. In the results of the research, the authors did not take into account publications and data from Russia and Ukraine. In particular, Osphya bipunctata (Fabricius, 1775) lives in these countries and information about the area is available on the platform https://www.gbif.org/species/4453653. See publications (https://doi.org/10.3390/d14121128; https://doi.org/10.3390/d14100825). You need to add these publications to the references.

Response: Thanks for your rigorous comments. We have added relevant data and cited the publications mentioned above. Please review the details in the revision manuscript.

Q: 4. Some captions to the drawings have been moved to other pages, which makes it difficult to read the manuscript.

Response: Thanks for your rigorous comments, and accordingly we have revised it to make read better.

Q: 5. In the discussion, due to the lack of all publications for analysis, the authors are mistaken in their conclusions. This requires making changes to the text.

Response: Thank you for your comments. We supplemented the geographic data from the above relevant publications, and conducted the analyses again and updated all figures. Generally, there is little change in the results, so we believe that our conclusions are quite objective. No matter how, thanks for your providing information.

Round 2

Reviewer 3 Report

Dear authors. Thanks for the answers and corrections in the manuscript.